# Structural, Electronic and Optical Properties of Some New Trilayer Van de Waals Heterostructures

**DOI:** 10.3390/nano13091574

**Published:** 2023-05-08

**Authors:** Beitong Cheng, Yong Zhou, Ruomei Jiang, Xule Wang, Shuai Huang, Xingyong Huang, Wei Zhang, Qian Dai, Liujiang Zhou, Pengfei Lu, Hai-Zhi Song

**Affiliations:** 1Quantum Research Center, Southwest Institute of Technical Physics, Chengdu 610041, China; beitong20@163.com (B.C.);; 2School of Electronic Engineering, Chengdu Technological University, Chengdu 611730, China; 3Faculty of Science, Yibin University, Yibin 644007, China; 4Institute of Fundamental and Frontier Sciences, University of Electronic Science and Technology of China, Chengdu 610054, China; 5State Key Laboratory of Information Photonics and Optical Communications, Beijing University of Posts and Telecommunications, Beijing 100876, China; 6State Key Laboratory of High Power Semiconductor Lasers, Changchun University of Science and Technology, Changchun 130013, China

**Keywords:** two-dimensional material, van der Waals heterostructure, trilayer, the first-principles calculation

## Abstract

Constructing two-dimensional (2D) van der Waals (vdW) heterostructures is an effective strategy for tuning and improving the characters of 2D-material-based devices. Four trilayer vdW heterostructures, BP/BP/MoS_2_, BlueP/BlueP/MoS_2_, BP/graphene/MoS_2_ and BlueP/graphene/MoS_2_, were designed and simulated using the first-principles calculation. Structural stabilities were confirmed for all these heterostructures, indicating their feasibility in fabrication. BP/BP/MoS_2_ and BlueP/BlueP/MoS_2_ lowered the bandgaps further, making them suitable for a greater range of applications, with respect to the bilayers BP/MoS_2_ and BlueP/MoS_2_, respectively. Their absorption coefficients were remarkably improved in a wide spectrum, suggesting the better performance of photodetectors working in a wide spectrum from mid-wave (short-wave) infrared to violet. In contrast, the bandgaps in BP/graphene/MoS_2_ and BlueP/graphene/MoS_2_ were mostly enlarged, with a specific opening of the graphene bandgap in BP/graphene/MoS_2_, 0.051 eV, which is much larger than usual and beneficial for optoelectronic applications. Accompanying these bandgap increases, BP/graphene/MoS_2_ and BlueP/graphene/MoS_2_ exhibit absorption enhancement in the whole infrared, visible to deep ultraviolet or solar blind ultraviolet ranges, implying that these asymmetrically graphene-sandwiched heterostructures are more suitable as graphene-based 2D optoelectronic devices. The proposed 2D trilayer vdW heterostructures are prospective new optoelectronic devices, possessing higher performance than currently available devices.

## 1. Introduction

Van der Waals (vdW) heterostructures [1,2,3] containing different two-dimensional (2D) material monolayers have recently attracted interest for fundamental physics in low-dimensional systems [4,5,6,7,8] and for applications in microelectronic and optoelectronic devices [9,10,11,12]. Several kinds of typical 2D materials such as graphene [13,14,15], transition metal dichalcogenide (TMD) [16,17] and phosphorene [18,19,20] have been incorporated into this construction. In graphene/MoS_2_, good thermal stability with high atom cohesive energy was confirmed [21] and found to mainly be associated with vdW coupling [22], and an electric field applied at the hetero-interface can control the Schottky barriers and Ohmic contacts [23]. Bilayer graphene/black-phosphorous (BP) was proposed to protect BP from its structural and chemical degradation with a slightly opened graphene band gap while still maintaining the electronic characteristics of graphene and BP monolayers [24,25]. Graphene/hexagonal-boron-nitride (hBN) appears to have good electric-field-driven switching, which is beneficial to the field-effect transistor [26]. MoS_2_/MX_2_ (M = Mo, Cr, W; X = S, Se) bilayer heterostructures exhibit semiconductor properties with an indirect band gap, except for MoS_2_/WSe_2_ [27], and all of them can undergo transiting from semiconductors to metals with the strain and band gap decreasing with the electric field [28], showing prospects for construction of ultrathin flexible devices and for application in viable optoelectronic fields. The TMD/BN heterostructures were explored for application in photocatalytic fields owing to the direct bandgap and powerful built-in electric field across the interface [29]. The type-II band alignments of MoS_2_/ZnO and WS_2_/ZnO help to separate the photo-generated charge effectively, and all TMD/graphene-like-ZnO structures show optical absorption in the visible and infrared regions [30]. Blue-phosphorous (BlueP)/BP possesses a tunable bandgap, band edges and electron-hole behavior by applying perpendicular electric field, and it thus shows potential for use in novel optoelectronic devices [31]. BlueP/GaN vdW heterostructure was found active for facilitating charge injection and thus promising for unipolar electronic device applications [32]. The photoelectric performances of BP/TMD heterostructures can be improved by applying compressive stress [33] or an electric field [34]. BP/MoS_2_ demonstrated an improved response and absorption characteristics in photodetectors [35]. Many BP/XT_2_ (X = Mo, W; T = S, Se, Te) heterostructures with type-II band alignments would be prospective as spin-filter devices [36]. BlueP/MoS_2_ has been considered the next generation of photovoltaic devices and water-splitting materials due to its wide optical response range and good light absorption ability [37]. In addition to the theoretical research such as that mentioned above, there have been experimental studies on bilayer vdW heterostructures to explore their practical prospects. The BP/MoS_2_ heterostructure photodetector was prepared and demonstrated with a wide range of current-rectifying behavior, microsecond response speed and high detectivity [38]. Graphene/BP heterostructure phototransistors exhibited good photoconductive gain and thus an ultrahigh photoresponse at near-infrared wavelength, implying the potential applications in remote sensing and environmental monitoring [39]. It is apparent that building vdW heterostructures is an effective way to achieve higher performance from devices based on graphene, phosphorenes and TMDs.

As an extension to the above idea, trilayer vdW heterostructures have recently been attracting more and more attention, since one more layer might open more space for tuning the properties further [40,41]. Simulation by Datta et al. demonstrated that the electronic properties of MoS_2_/MX_2_/MoS_2_ exhibit a smaller electron effective mass and a semiconductor-to-metal transition under tensile strain [42]. Device level performance of MoS_2_/MX_2_/MoS_2_ were further calculated, showing that these trilayer heterostructures would be prospective to construct highly sensitive field effect transistors for nano-biomolecules sensing as well as for pH sensing [43]. Bafekry et al. theoretically analyzed trilayer vdW heterostructures in which MoS_2_ monolayer is encapsulated by two graphitic carbon nitride monolayers, revealing their magnetic metallic characteristics [44] probably applicable in nano-spintronic devices. Liu et al. proposed trilayer heterostructures MoS_2_/SiC/MoS_2_ and SiC/MoS_2_/SiC, and predicted their strain-induced tunable band structure promising for future optoelectronic devices [45]. Calculation on BlueP/MoX_2_-based trilayer heterostructures demonstrated the excellent ability to accelerate the separation of photo-generated electron-hole pairs and the controllable power conversion efficiency indicating great potential for photocatalysis and photovoltaics [46]. BN/graphene/BN was proposed and simulated to open the graphene bandgap and enhance the application potential for radio-frequency devices and switching transistors [47,48]. Xu et al. theoretically argued that MoS_2_-graphene-based trilayer heterostructures are of unique optoelectronic properties tunable by external electric field and thus of great potential for solar energy harvesting and conversion [49]. External electric field modulation effects were also simulated on PtSe_2_/graphene/graphene and graphene/PtSe_2_/graphene, revealing their usefulness to guide the design of high-performance field effect transistors [50]. In an experimental work, Long et al. fabricated WSe_2_/graphene/MoS_2_ p-g-n heterostructure photodetector, which achieved a wide detection wavelength range of 400–2400 nm and a short rise/fall time [51]. Li et al. fabricated photodetectors based on dielectric shielded MoTe_2_/graphene/SnS_2_ p–g–n junctions, and achieved extraordinary responsivity as high as 2600 A/W, detectivity as good as 10^13^ Jones with fast photoresponse in the range of 405–1550 nm [52]. It is thus clear that constructing trilayer vdW heterostructures can improve further the performance of electronic and optoelectronic devices based on graphene, phosphorenes and TMDs. However, the reported trilayer heterostructures used mainly symmetrical structures and/or contained two types of materials. Further efforts should be made to construct new vdW heterostructures.

In this work, we attempt to construct and study trilayer vdW heterostructures with asymmetric structures and/or three different types of 2D materials. Four kinds of trilayer heterostructures containing MoS_2_, phosphorene and/or graphene are simulated in terms of structural, optical and optoelectronic properties. It is suggested that sophisticated 2D trilayer vdW heterostructures can provide further optimized characters for a new generation of optoelectronic devices. 

## 2. Calculation Methods

In this study, we designed four trilayer vdW heterostructures BP/BP/MoS_2_, BlueP/BlueP/MoS_2_, BP/graphene/MoS_2_ and BlueP/graphene/MoS_2_, which are structurally asymmetric and/or composed of more than two different types of 2D materials. The first-principles calculation was performed using the tool of Vienna ab initio simulation package (VASP) [53] within Density Functional Theory (DFT) and the projector-augmented wave (PAW) method [54]. The generalized gradient approximation (GGA) with the Perdew–Burke–Ernzerhof (PBE) method was applied for calculating electron exchange and correlation potentials [54,55]. Due to the weak vdW interaction between the monolayers, the opt-PBE vdW method [55,56] was adopted to modify the vdW dispersion. To avoid the interaction between adjacent slab models, the thickness of the vacuum layer was set to 20 Å. The energy cutoff, convergence criteria of total energy and force were set to 450 eV, 10^−5^ eV and 0.01 eV/A, respectively. When optimizing the trilayer atomic structures, 1 × 7 × 1, 3 × 3 × 1, 2 × 2 × 1 and 3 × 3 × 1 k-point sets were adopted for BP/BP/MoS_2_, BlueP/BlueP/MoS_2_, BP/graphene/MoS_2_ and BlueP/graphene/MoS_2_, respectively. When calculating the electronic structures, 5 × 21 × 1, 16 × 16 × 1, 5 × 5 × 1 and 8 × 8 × 1 k-point sets were adopted for the four structures, respectively. When calculating the optical properties, 8 × 21 × 1, 20 × 20 × 1, 10 × 10 × 1 and 13 × 13 × 1 k-point sets were adopted for the four structures, respectively. The above-mentioned k-point sets were actually determined by convergence tests: increasing the k-point set scale until a state in which the calculation results do not change beyond the convergence standard any longer. It reflects that calculations for structural, electronic and optical properties need more and more computational cost and accuracy.

To establish the stability of the proposed vdW heterostructures, we should in principle evaluate their vibrational frequencies to seek real and positive phonon parameters. Nevertheless, most of the first-principles simulations on vdW heterostructures took the binding energy as the figure of merit to characterize the structure stability [21,27,30,36], because the vibrations are so computationally expensive that a complex structure would be technically hard to deal with. In our case, the proposed atomic structures are even larger and complicated, so we judge the structure stability and thus the experimental feasibility by calculating the binding energy, which would be negative for a stable system [23]. The binding energy (*E*_b_) is calculated using the following equation: (1)$$E_{b}=E_{hts}-E_{A}-E_{B}-E_{C}$$
where *E*_hts_ represents the overall energy of the heterostructure and *E*_A_, *E*_B_, and *E*_C_ represent the energy of each monolayer in its free state. The work function, used as the key parameter to deduce the band alignment, is described by
(2)$$W=E_{vac}-E_{f}$$
where *E*_vac_ is the energy of a stationary electron in the vacuum and *E*_f_ is the Fermi level.

The macroscopic optical response function is usually measured by the complex dielectric function $\varepsilon (\omega )=\varepsilon_{1}(\omega )+i\varepsilon_{2}(\omega )$, where *ω* represents the circular frequency of light. The real part $\varepsilon_{1}(\omega )$ can be processed via the Kramers–Kronig relation and the imaginary part $\varepsilon_{2}(\omega )$ is produced by Fermi’s golden rule [57]. Based on these parameters, we can obtain the absorption coefficient by [58],
(3)$$\alpha (\omega )=\frac{\sqrt{2}\omega }{c}{\left\{{\left[{\varepsilon_{1}}^{2}(\omega )+{\varepsilon_{2}}^{2}(\omega )\right]}^{1/2}-\varepsilon_{1}(\omega )\right\}}^{1/2}$$

## 3. Results and Discussion

### 3.1. Structural Parameters

Due to the surface relaxation and interfacial mismatch, the trilayer vdW heterostructure supercells must be carefully set, for which we here extend monolayer cells individually to satisfy the length matching between different layers. The BP/BP/MoS_2_ heterostructure is constructed from a $4\sqrt{3}$ × 1 supercell MoS_2_ monolayer and a 5 × 1 supercell bilayer-BP, while the BlueP/BlueP/MoS_2_ heterostructure is formed by stacking a 3 × 3 supercell bilayer-BlueP on a 3 × 3 supercell MoS_2_ monolayer. After structure optimization calculation, we obtain the rectangular lattice constants *a* = 22.227 and *b* = 3.278 Å for BP/BP/MoS_2_, and the hexagonal lattice constants *a* = *b* = 9.642 Å for BlueP/BlueP/MoS_2_. For BP/graphene/MoS_2_, bilayer BP/graphene cells are first constructed by matching 3 × 1 BP with $4\sqrt{3}$ × 1 supercell graphene, and then the 1 × 2 supercell of the bilayer BP/graphene is set to match the 2$\sqrt{3}$ × 3 supercell MoS_2_. For BlueP/graphene/MoS_2_, a 3 × 3 supercell monolayer BlueP, 4 × 4 supercell graphene, and 3 × 3 supercell monolayer MoS_2_ are successively stacked. The optimized rectangular lattice constants of BP/graphene/MoS_2_ are *a* = 10.256 Å and *b* = 8.907 Å, and the hexagonal ones of BlueP/graphene/MoS_2_ are *a* = *b* = 9.776 Å. The simulated models of BP/BP/MoS_2_, BlueP/BlueP/MoS_2_, BP/graphene/MoS_2_ and BlueP/graphene/MoS_2_ are depicted in Figure 1.

The trilayer vdW heterostructures will have different energies with different layer spacing. Figure 1 gives the layer spacing when the system reaches the final equilibrium. The calculated binding energy of the four trilayer heterostructures is −21.12, −65.41, −21.92 and −65.78 eV, respectively. The negative binding energies indicate that all the four proposed vdW heterostructures are structurally highly stable like the reported bilayer heterostructures [21,36]. The structure stability may imply the fabrication feasibility to some degree. As is known, BP/MoS_2_ [38], graphene/MoS_2_ [59], graphene/BP [39] bilayer heterostructures, and TMD/graphene/MoS_2_ [51] trilayer heterostructures have been successfully fabricated using physical and chemical methods including mechanical stripping, liquid-phase stripping, hydrothermal and chemical vapor deposition. BlueP 2D monolayer has been well fabricated by molecular beam epitaxy [60], and the possibility to form its heterostructure with other 2D materials has been stated by a few studies [37,46]. Combining together the above fabrication abilities and considering the confirmed structure stability, the preparation of our proposed vdW heterostructures will be experimentally feasible. Noting the binding energy differences among these heterostructures, BlueP/BlueP/MoS_2_ and BlueP/graphene/MoS_2_ might be easier to be fabricated than BP/BP/MoS_2_ and BP/graphene/MoS_2_, although BlueP-related 2D materials seem currently farther away from practical manufacture than BP-related ones.

### 3.2. Electronic Properties

#### 3.2.1. Band Structure

Figure 2a,b demonstrate the projected band structures of BP/BP/MoS_2_ and BlueP/BlueP/MoS_2_ calculated using the PBE method. BP/BP/MoS_2_ exhibits the conduction band minimum (CBM) located between the Y and G points and mostly contributed by MoS_2_ (red lines) as well as the valance band maximum (VBM) located at the high-symmetry G point and mostly contributed by BP (blue lines). These are qualitatively consistent with our calculation of the bilayer counterpart BP/MoS_2_ (in agreement with [34]), suggesting that the high hole mobility of BP and high electron mobility of MoS_2_ are still preserved. The indirect bandgap of the $4\sqrt{3}$ × 1 supercell MoS_2_, 1.338 eV, and the direct bandgap of the 5 × 1 supercell bilayer-BP, 0.706 eV, are reduced compared with those in bilayer BP/MoS_2_ (MoS_2_ 1.398 eV, BP 0.876 eV), which are already lower than the free monolayer ones (MoS_2_ 1.69 eV, BP 0.9 eV [36]). Significantly, the whole BP/BP/MoS_2_ heterostructure possesses an indirect band gap of 0.167 eV, nearly half that of bilayer BP/MoS_2_, 0.326 eV. It is seen that this trilayer heterostructure effectively tunes the bandgap further. For BlueP/BlueP/MoS_2_, the CBM is located between the G and M points and is also mostly contributed by MoS_2_ (red lines), whereas the VBM is located at the high-symmetry K point and is mostly contributed by BlueP (blue lines), consistent with the calculated results of the bilayer counterpart BlueP/MoS_2_ (in agreement with [37]). Similar to the above BP/BP/MoS_2_, the direct bandgap of the 3 × 3 supercell MoS_2_ monolayer, 1.298 eV, and the indirect bandgap of the 3 × 3 supercell BlueP monolayer, 1.624 eV, are smaller than those in the bilayer BlueP/MoS_2_ (MoS_2_ 1.328 eV, BlueP 1.698 eV), which are already decreased compared with the corresponding free monolayer ones (MoS_2_ 1.69 eV, BlueP 1.908 eV [2,37]). The whole trilayer heterostructure possesses an indirect band gap of 1.2632 eV, smaller than that of the bilayer BlueP/MoS_2_, 1.3746 eV. It falls in between the BlueP/MoS_2_/BlueP of 1.013 eV and MoS_2_/BlueP/MoS_2_ of 1.561 eV reported by Han et al. [46], again implying the further effective tuning by such an asymmetric trilayer heterostructure. The BP(BlueP) and MoS_2_ bandgap decrease as they compose bilayer or trilayer heterostructures, which can be attributed to ~2.5% and ~2% lattice mismatch and a weak vdW force [35]. A whole-bandgap decrease with the addition of a BP(BlueP) monolayer was discovered by Dong et al. and Qiao et al. in their studies on BP [57,61]; bandgap engineering by means of the simple addition of monolayers appears thus to be effective. Moreover, it is interesting here that the VBM of BlueP and MoS_2_ are becoming much closer, which supports application with the function of carrier transferring or tunneling. 

Figure 2c,d show the projected band structure of BP/graphene/MoS_2_ and BlueP/graphene/MoS_2_ trilayer vdW heterostructures calculated using the PBE method. What is immediately significant is that the BP/graphene/MoS_2_ structure opens the bandgap of graphene to 0.051 eV, which is over one order of magnitude larger than the opening by the reported bilayer heterostructures such as graphene/BP (~0.0013 eV) [24], graphene/ZnO (~0.0057 eV) [30], and graphene/MoS_2_ (~0.0007 eV) [49], and remarkably larger than those by other trilayer heterostructures such as graphene/ZnO/MoS_2_ (~0.0048 eV) [30], graphene/nitrogene/graphene (~0.04 eV) [40], and graphene/BN/graphene (~0.020 eV) [47]. BlueP/graphene/MoS_2_ opens the bandgap of graphene by 0.0093 eV, similar to many other heterostructures. The effectiveness of the graphene bandgap opening in BP/graphene/MoS_2_ can hardly be attributed to strain, because the strains of graphene in both structures are close to each other (3.2% and 2.8%) and close to those of the reported structures such as graphene/BP (3%) [24]. We believe it originates from the united vdW coupling forces among BP and MoS_2_ monolayers and possible built-in electric fields. The remarkable graphene bandgap opening suggests potential applications in temperature-sensitive devices [62]. On the basis of guaranteeing the excellent electronic properties of graphene, it can also be applied to field effect transistors with a high switching ratio. It is worth noting that these band gap values are grossly underestimated owing to adopting PBE functional calculation, so the real bandgaps of graphene in such heterostructures may be larger to match more extended application prospects. Furthermore, the band gap values of BP (~1.018 eV) and BlueP (~1.998 eV) in these two trilayer heterostructures are, qualitatively as graphene does, slightly larger than the free monolayers (BP of 0.9 eV [36] and BlueP of 1.908 [37]), which is in contrast to the first two structures BP/BP/MoS_2_ and BlueP/BlueP/MoS_2_. There seem to be two types of bandgap engineering mechanisms in trilayer vdW heterostructures, providing wider space for novel device design.

We also calculated the local density of state (LDOS) and total density of states (TDOS) of BP/BP/MoS_2_ and BlueP/BlueP/MoS_2_, as shown in Figure 3. The CBM state is mainly contributed by the Mo element and the VBM state relies on the P element, but the small difference between LDOS and TDOS may suggest that the interlayer vdW interaction also has a role in tuning the energy bands. What is quite significant is that only one more monolayer of BP(BlueP) increases the DOS to more than twice compared with the corresponding bilayer heterostructures. A similar phenomenon occurs in BP/graphene/MoS_2_ and BlueP/graphene/MoS_2_, where the DOS also increases due to the insertion of a single graphene monolayer with respect to BP/MoS_2_ and BlueP/MoS_2_ bilayer heterostructures. It evidences the effectiveness of the proposed trilayer vdW heterostructures in optimizing the performance of 2D material systems.

#### 3.2.2. Band Alignment

The band alignment of the vdW heterostructure is important in the design of new 2D-material structures. It is thus necessary to specifically pick up the band alignment data from the proceeding band structures with the help of calculating the work function. In Figure 4, for the BP/BP/MoS_2_ and BlueP/BlueP/MoS_2_ heterostructures, p-type bilayer-BP(BlueP) (*E_f_*closer to VBM) and n-type MoS_2_ (*E_f_* closer to CBM) form type-II band alignment, as is expected. This band alignment has carrier confinement effects [63], favoring the electron-hole separation and then increasing the internal gain and responsivity of photodetectors. In brief, BP/BP/MoS_2_ shows a more obvious type-II band alignment with a larger offset on the valence band and will produce a stronger carrier confinement effect compared with BP/MoS_2_ [37,40]. With graphene inserted, BP/graphene/MoS_2_ and BlueP/graphene/MoS_2_ heterostructures naturally show a type-I quantum-well-like band alignment, providing the potential to construct nano-scaled bipolar transistors, field-effect transistors and photon transistors, which may need three different energy structures. In contrast to the Schottky contact of graphene/MoS_2_ and graphene/BP(BlueP), here, semiconductor heterojunctions are formed. By applying an electric field across this semiconductor heterojunction, the graphene bandgap can be further flexibly tuned to exhibit different band alignments, which is of significance for the construction of new types of field-effect photodetectors [49].

### 3.3. Optical Absorption Spectra

On the optical absorption properties of 2D materials, many studies have been devoted to monolayer and bilayer vdW heterostructures, but little research has been carried out on trilayer ones, let alone a comparison between the bilayer and trilayer heterostructures [48]. The absorption spectra of BP/BP/MoS_2_ and BlueP/BlueP/MoS_2_ trilayer heterostructures are shown in Figure 5a,b. The absorption edge redshifts effectively expand the application ranges from covering the short-wave infrared band to covering the mid-wave infrared band for BP-based material, and from covering the visible band to covering the short-wave infrared band for BlueP-based material. This change is directly associated with the shrinkage of the bandgaps observed in Figure 3, but there seems to be something more incorporated as we see the following. Obviously, with only one more phosphorene monolayer, i.e., ~1/3 increase in material quantity, the spectral absorption intensity increases to be more or less twice with respect to monolayer and bilayer counterparts. This is more than expected, meaning that these two trilayer heterostructures greatly improve the quantum efficiency in optoelectronic devices [64] in wide spectral ranges. This sort of enhancement might be the result of complex interlayer vdW interactions. In more detail, the absorption of BP/BP/MoS_2_ is much stronger (reaching 10^5^ cm^−1^) than that of BlueP/BlueP/MoS_2_ in the near-infrared range, but the latter displays more significant absorption in the ultraviolet range. They can thus be applied effectively in different scenes. As a whole, these results make up for the shortage of monolayer 2D materials in optical characteristics and are beneficial for optoelectronic devices with the need for light absorption enhancement and absorption range broadening.

Figure 6 displays the absorption coefficients of bilayer heterostructures graphene/BP, graphene/BlueP and graphene/MoS_2_, and the trilayer heterostructures BP/graphene/MoS_2_ and BlueP/graphene/MoS_2_. The inset shows that the two kinds of trilayer heterostructures have stronger light absorption in the range below 0.8 eV than the related bilayer heterostructures, so they are better choices for graphene-based 2D materials to realize photodetectors widely responding to short-, mid- and long-wave infrared light. In the range of 2.2–6 eV, BP/graphene/MoS_2_ shows remarkably higher light absorption than the related bilayer heterostructures, making it able to realize better graphene-based optoelectronic devices working in a wide range from visible to deep ultraviolet. BlueP/graphene/MoS_2_ shows better light absorption characteristics than the related bilayer heterostructures in the range of 4–6 eV, so it is more suitable for use as a graphene-based solar-blind ultraviolet photodetector. These asymmetric graphene-sandwiched vdW heterostructrues are thus proved to be effective in tuning and improving the performance of graphene-based 2D materials on the basis of guaranteeing the excellent electronic properties of graphene.

## 4. Conclusions

Using the first-principles calculation based on DFT theory, we proposed and simulated trilayer 2D vdW heterostructures: BP/BP/MoS_2_, BlueP/BlueP/MoS_2_, BP/graphene/MoS_2_ and BlueP/graphene/MoS_2_. The atomic structures of these heterostructures are all stable and look feasible to prepare. For BP/BP/MoS_2_ and BlueP/BlueP/MoS_2_, all the bandgaps were tuned to be smaller, to effectively compensate for the shortage of corresponding monolayers and to satisfy a wider spectrum of applications with respect to the bilayer structures BP/MoS_2_ and BlueP/MoS_2_, respectively. Their higher DOS, much stronger light absorption and more obvious type-II band alignment will realize better performance of optoelectronic devices working in a wide spectrum from mid-wave (short-wave) infrared to violet. In contrast, for BP/graphene/MoS_2_ and BlueP/graphene/MoS_2_, most of the bandgaps were enlarged, implying different tuning mechanisms related to vdW interaction. Specifically in BP/graphene/MoS_2_, the graphene bandgap was well opened to 0.051 eV, more than one order of magnitude larger than usual, indicating more space for graphene tuning in trilayer vdW heterostructures. Accompanying the bandgap increase, both BP/graphene/MoS_2_ and BlueP/graphene/MoS_2_ exhibit absorption enhancement in wide spectral ranges, suggesting that these trilayer vdW heterostructures show better prospects as graphene-based 2D material optoelectronic devices with broad responses in the short- to long-wave infrared, visible to deep ultraviolet and solar-blind ultraviolet bands. It is concluded that the proposed 2D trilayer vdW heterostructures are potentially applicable as novel optoelectronic devices.

## Figures and Tables

**Figure 1 nanomaterials-13-01574-f001:**
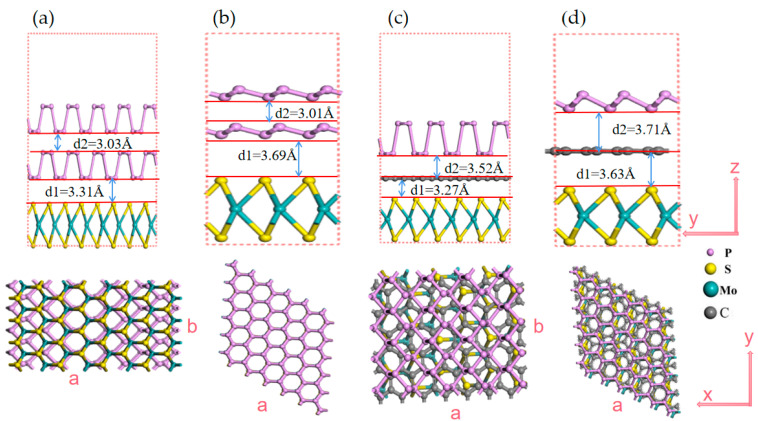
The models of trilayer 2D vdW heterostructures. (**a**) BP/BP/MoS_2_; (**b**) BlueP/BlueP/MoS_2_; (**c**) BP/graphene/MoS_2_; (**d**) BlueP/graphene/MoS_2_.

**Figure 2 nanomaterials-13-01574-f002:**
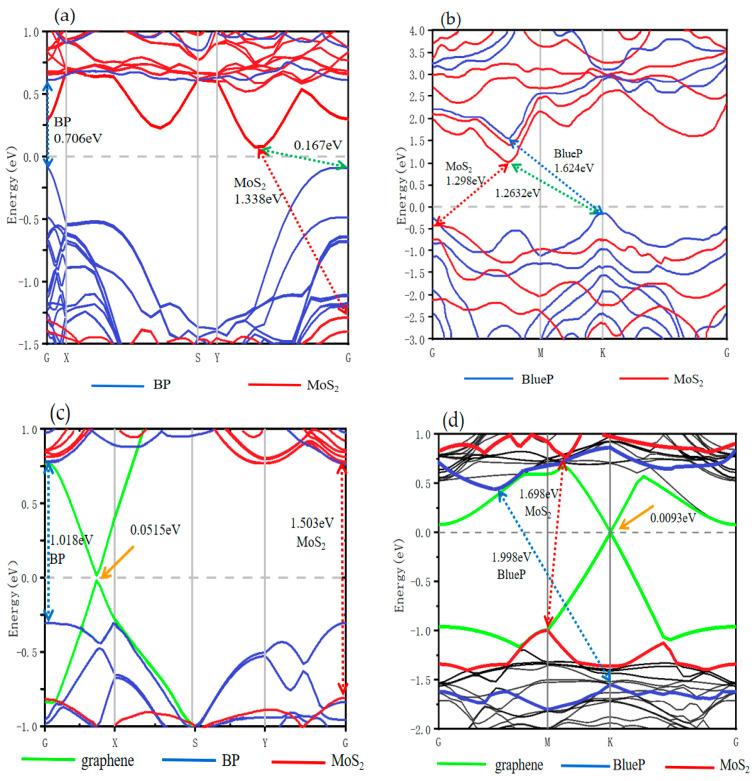
Calculated projected band structures for trilayer vdW heterostructures (**a**) BP/BP/MoS_2_; (**b**) BlueP/BlueP/MoS_2_; (**c**) BP/graphene/MoS_2_; (**d**) BlueP/graphene/MoS_2_. The Fermi level *E_f_* is set as zero energy.

**Figure 3 nanomaterials-13-01574-f003:**
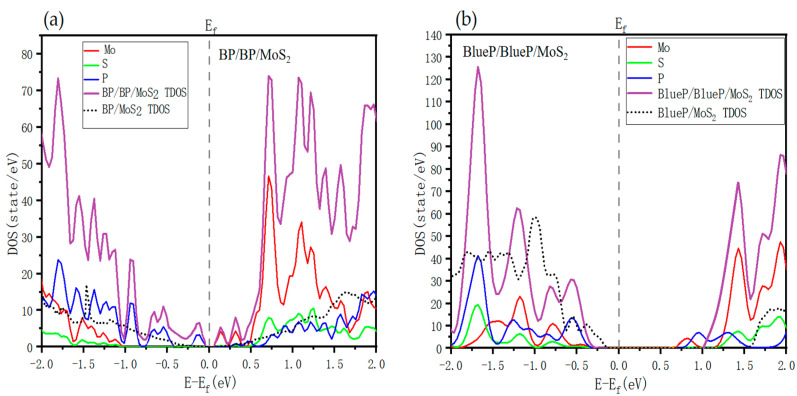
Calculated LDOS and TDOS for (**a**) BP**/**BP/MoS_2_ and (**b**) BlueP/BlueP/MoS_2_. The different colors represent different elements. The gray dashed line indicates Fermi energy. That of bilayer counterpart is also shown for comparison.

**Figure 4 nanomaterials-13-01574-f004:**
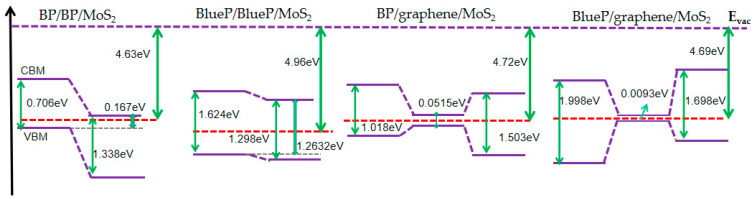
Band alignments of BP/BP/MoS_2_, BlueP/BlueP/MoS_2_, BP/graphene/MoS_2_ and BlueP/graphene/MoS_2_. The red dashed lines represent the Fermi levels. The vacuum level (*E*_vac_) is shown by a purple dashed line.

**Figure 5 nanomaterials-13-01574-f005:**
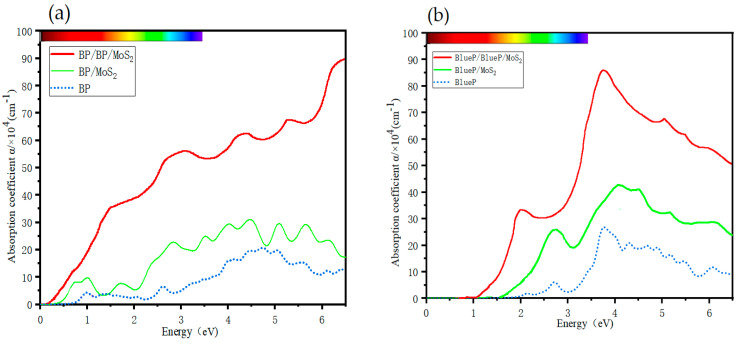
Optical absorption coefficient of (**a**) monolayer BP and MoS_2_, bilayer BP, BP/MoS_2_ and BP/BP/MoS_2_; and (**b**) monolayer BlueP and MoS_2_, bilayer BlueP, BP/MoS_2_ and BlueP/BlueP/MoS_2_.

**Figure 6 nanomaterials-13-01574-f006:**
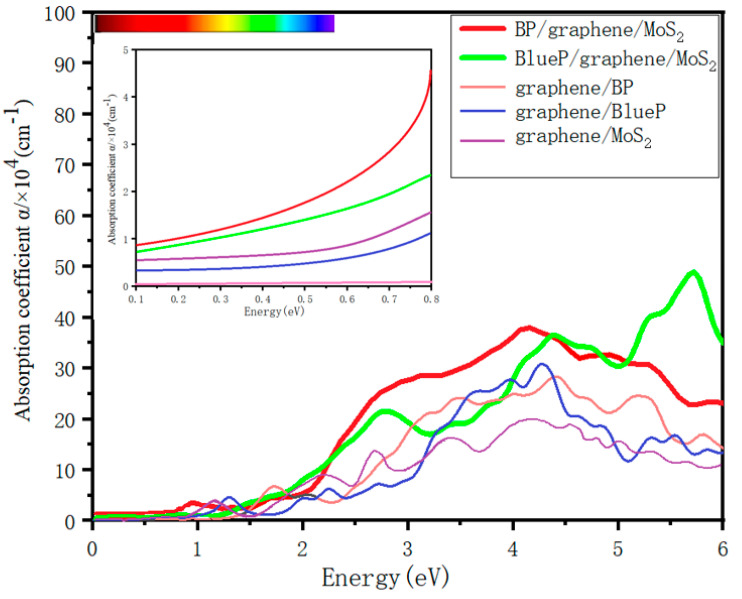
Optical absorption coefficient of BP/graphene/MoS_2_ and BlueP/graphene/MoS_2_ heterostructures, and the optical absorption coefficient of graphene/BP, graphene/BlueP and graphene/MoS_2_ are also shown for comparison.

## Data Availability

The data presented in this study are available on request from the corresponding author.

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
