# Peer review of "Structural, Electronic and Optical Properties of Some New Trilayer Van de Waals Heterostructures"

_nanomaterials, 2023, doi:10.3390/nano13091574_

Round 1

Reviewer 1 Report

This is a well written manuscript and represents interesting results. The language requires minor editing. My main concern is about the practicable aspect of this work. In a gist, the authors have designed new materials on VASP, and modeled their electronic responses. However, there's no clear indication that there is any experimental basis for this. If there isn't then there is no indication that these materials can even be synthesized and realized. Without this insight it is almost impossible to comment on the impact of this work. The authors need to address this issue.

Minor editing work is needed. This can be easily done by working with a professional editor/proofreader.

Author Response

Dear Reviewer,

Thank you very much for your review and comments on our manuscript. We would like here to respond to your opinions.

About the English writing, we have asked the editing services (https://www.mdpi.com/ authors/english  for help. The changes are all clearly shown by revision tracking function in the revised manuscript. We hope that the editing service satisfies this requirement.

About the practical aspect of our work, it is true that there is not exact experimental basis yet because what we proposed are new structures nobody has ever tried. But one can predict the experimental feasibility by combining the currently available techniques. For this improvement, we insert some related previous experimental reports in the introduction section to evidence the experimental basis for fabricating our structures, and add several sentences analyzing the preparation feasibility in the subsection “3.1. Structural parameters”.

In details, the revisions include:

“In addition to the theoretical research like mentioned above, there have been experimental studies on bilayer vdW heterostructures to explore their practical prospects. The BP/MoS2 heterostructure photodetector were prepared and demonstrated with a wide range of current-rectifying behavior, microsecond response speed and high detectivity [38]. Graphene/BP heterostructure phototransistors exhibited good photoconductive gain and thus an ultrahigh photoresponse at near-infrared wavelength, implying the potential applications in remote sensing and environmental monitoring [39].” , inserted to line 70-77, page 2 in the revised manuscript;

“In an experimental work, Long et al. fabricated WSe2/graphene/MoS2 p-g-n hetero-structure photodetector, which achieved a wide detection wavelength range of 400 - 2400 nm and a short rise/fall time [51]. Li et al. fabricated photodetectors based on di-electric shielded MoTe2/graphene/SnS2 p–g–n junctions, and achieved extraordinary responsivity as high as 2600 A/W, detectivity as good as 1013 Jones with fast photoresponse in the range of 405-1550 nm [52].”, inserted to lines 104-110, page 3 in the revised manuscript;

“The structure stability may imply the fabrication feasibility to some degree. As is known, BP/MoS2 [38], graphene/MoS2 [59], graphene/BP [39] bilayer heterostructures, and TMD/graphene/MoS2 [51] trilayer heterostructures have been successfully fabricated using physical and chemistry methods including mechanical stripping, liquid-phase stripping, hydrothermal and chemical vapor deposition. BlueP 2D monolayer has been well fabricated by molecular beam epitaxy [60], and the possibility to form its heterostructure with other 2D materials has been stated by a few studies [37,46]. Combining together the above fabrication abilities and considering the confirmed structure stability, the preparation of our proposed vdW heterostructures will be experimentally feasible. Noting the binding energy differences among these heterostructures, BlueP/BlueP/MoS2 and BlueP/graphene/MoS2 might be easier to be fabricated than BP/BP/MoS2 and BP/graphene/MoS2, although BlueP-related 2D materials seem currently farther away from practical manufacture than BP-related ones.”, inserted to lines 187-199, pages 4 – 5 in the revised manuscript to replace the original simple sentence “It confirms that all the proposed vdW heterostructures can be practically prepared and will be structurally stable.”.

The above arguments also serve as the responses to most of the reviewers’ evaluations marked with “must be improved”. But that for the question “Does the introduction provide sufficient background and include all relevant references?” will be addressed as follows. In order to provide sufficient background and include all relevant references, we append more literatures ([32, 43-45, 50, 60]) and more contents to describe the overall background as sufficiently as possible, into the first and second paragraphs. In detail, lines 52-55 are revised to be “… all of them can undergo transiting from semiconductors to metals with the strain and band gap decreasing with the electric field [28], showing prospects for construction of ultrathin flexible devices and for application in viable optoelectronic fields.”; lines 60-64 are revised to be “Blue-phosphorous (BlueP)/BP possesses a tunable bandgap, band edges and electron–-hole behavior by applying perpendicular electric field, and it thus shows potential for use in novel optoelectronic devices [31]. BlueP/GaN vdW heterostructure was found active for facilitating charge injection and thus promising for unipolar electronic de-vice applications [32].”; in lines 70-77, there inserted “Blue-phosphorous (BlueP)/BP possesses a tunable bandgap, band edges and electron–hole behavior by applying perpendicular electric field, and it thus shows potential for use in novel optoelectronic devices [31]. BlueP/GaN vdW heterostructure was found active for facilitating charge injection and thus promising for unipolar electronic device applications [32].”; in lines 86-97, there inserted “Device level performance of MoS2/MX2/MoS2 were further calculated, showing that these trilayer heterostructures would be prospective to construct highly sensitive field effect transistors for nano-biomolecules sensing as well as for pH sensing [43]. Bafekry et al. theoretically analyzed trilayer vdW heterostructures in which MoS2 monolayer is encapsulated by two graphitic carbon nitride monolayers, revealing their magnetic metallic characteristics [44] probably applicable in nano-spintronic devices. Liu et al. proposed trilayer heterostructures MoS2/SiC/MoS2 and SiC/MoS2/SiC, and predicted their strain-induced tunable band structure promising for future optoelectronic devices [45]. Calculation on BlueP/MoX2-based trilayer heterostructures demonstrated the excellent ability to accelerate the separation of photo-generated electron-hole pairs and the controllable power conversion efficiency indicating great potential for photocatalysis and photovoltaics [46].”; in lines 99-112, there inserted “Xu et al. theoretically argued that MoS2-graphene-based trilayer heterostructures are of unique optoelectronic properties tunable by external electric field and thus of great potential for solar energy harvesting and conversion [49]. External electric field modulation effects were also simulated on PtSe2/graphene/graphene and graphene/PtSe2/graphene, revealing their usefulness to guide the design of high-performance field effect transistors [50]. In an experimental work, Long et al. fabricated WSe2/graphene/MoS2 p-g-n heterostructure photodetector, which achieved a wide detection wavelength range of 400 - 2400 nm and a short rise/fall time [51]. Li et al. fabricated photodetectors based on dielectric shielded MoTe2/graphene/SnS2 p–g–n junctions, and achieved extraordinary responsivity as high as 2600 A/W, detectivity as good as 1013 Jones with fast photoresponse in the range of 405-1550 nm [52]. It is thus clear that constructing trilayer vdW heterostructures can improve further the performance of electronic and optoelectronic devices based on graphene, phosphorenes and TMDs.”; the sentence “The heterostructure WSe2/graphene/MoS2 has achieved a wide detection wavelength range of 400-2400 nm and a short rise/fall time in experiments [39].” in lines 82-84 is omitted since it has been rewritten in the above.

   We hope these revisions can address what the reviewer strongly concerns.

Sincerely yours,

Hai-Zhi Song

Southwest Institute of Technical Physics

and

University of Electronic Science and Technology of China

Tel: +88-28-68180751

Email: hzsong1296@163.com,  hzsong@uestc.edu.cn

Reviewer 2 Report

In the present paper the authors have investigated four different trilayer 2D vdW heterostructures by ab initio calculations. 

The electronic structure, band alignment and optical absorption have been calculated, and applications are suggested as new optoelectronic devices possessing high performance.

The manuscript is of good interest for the Materials Science community.

The paper can be published once the following issues are addressed:

1) The authors stated that "the bandgap value of the BlueP/BlueP/MoS2 is very close to that of the traditional Si material (1.12 eV), so it can be an effective substitute for Si materials ...". However, the bandgap value of the BlueP/BlueP/MoS2 they have calculated within GGA approach is an underestimation the experimental value, therefore cannot be compared to the experimental bandgap of Si.

2) Do the authors have investigated the convergence vs the density of k-points of the bandgap opening up in the graphene heterostructures? What about the convergence vs the mesh of k-points of the optical absorption coefficient of BP/graphene/MoS2 and BlueP/graphene/MoS2 heterostructures?

3) In the methods section, the authors stated that they judge the experimental feasibility by calculating the binding energy, which would be negative for a stable structure. Really, that is not completely exact. Indeed, to establish the stability of the structures, the vibrational frequencies should be evaluated as well. A structure is stable when all the frequencies are real and positive. However, the vibrations are more computational expensive than electronic properties, so this calculation is usually not performed for complex structures such as those considered in this paper. In the manuscript, the heterostructures have been demonstrated to have negative binding energy, so in the case they are stable they will form bound systems.

As a minor comment, in the caption of figure 4: "Calculated TDOS and TDOS for ..." should be "Calculated LDOS and TDOS for ...".

Author Response

Dear Reviewer,

    Thank you very much for your review and comments on our manuscript. We would like here to respond to your opinions.

1) We appreciate very much the comment on whether the material bandgap actually matches Si or not. We agree that comparing GGA approach (underestimation the experimental we mentioned in our manuscript) with the experimental bandgap of Si is unreasonable. So we delete the sentence “Furthermore, the bandgap value of the BlueP/BlueP/MoS2 is very close to that of the traditional Si material (1.12 eV), so it can be an effective substitute for Si materials or a material matching Si in the application of micro- and nano-devices [9]", in line 239-241, page 6 in the revised manuscript.

2) Yes, we did investigate the convergence vs the density of k-points of not only the bandgap opening up in the graphene heterostructures but also many other parameters and characters. Some data about the convergence vs the mesh of k-points for the bandgap of BP/graphene/MoS2 are as follows:

         k-point set          bandgap (eV)    

      3×3×1              0.0098                 

      4×4×1              0.0215                 

      5×5×1              0.0513                  

      6×6×1              0.0512                 

      7×7×1              0.0512.

Accordingly, we chose 5×5×1 as the k-point set for calculating the electronic properties of BP/graphene/MoS2. For the optical property calculation, we attempted k-point sets changing from 7×7×1, 8×8×1 to 11×11×1, 13×13×1. We saw that, when the k points increase, the absorption curve goes up or down in different energy ranges, but from 10×10×1 on, it changes within 1% in all the energy ranges. This is the reason why we took 10×10×1 as the k-point set for the calculation of optical properties of BP/graphene/MoS2. Similar processes were also carried out on BlueP/graphene/MoS2, and then k-point sets 8×8×1 and 13×13×1 were determined for the calculation of electronic and optical properties of BlueP/graphene/MoS2, respectively.        

    To address this comment in the manuscript, there appended a few sentences as :

“When calculating the optical properties of the trilayer heterostructures, 8×21×1, 20×20×1, 10×10×1 and 13×13×1 k-point sets are adopted, respectively. The above mentioned k-point sets were actually determined by convergence tests: increasing the k-point set scale until a state in which the calculation results do not change beyond the convergence standard any longer. It reflects that calculations for structural, electronic and optical properties need more and more computational cost and accuracy.”, to lines 138-143, page 3 in the revised manuscript.

3) We agree that using binding energy to judge the experimental feasibility is not completely exact and that the vibrational frequencies should be evaluated to establish the structure stability. As is well argued by the reviewer, however, our complex structures are computationally so expensive in treating the vibrational frequencies that we decide to alternatively use binding energy. As a matter of fact, most reported first-principles studies on 2D heterostructures used binding energy to evaluate the structure stability. As to the experimental feasibility, it can be partly supported by this binding energy but it might need more support from the fabrication techniques. Therefore, we insert related explanations and revise somehow the original description for binding energy as:

“To establish the stability of the proposed vdW heterostructures, we should in principle evaluate their vibrational frequencies to seek real and positive phonon parameters. Nevertheless, most of the first-principles simulations on vdW heterostructures took the binding energy as the figure of merit to characterize the structure stability [21,27,30,36], because the vibrations are so computationally expensive that a complex structure would be technically hard to deal with. In our case, the proposed atomic structures are even larger and complicated, so we judge the structure stability and thus the experimental feasibility by calculating the binding energy, which would be negative for a stable system [23].” , to lines 144-153, page 3 in the revised manuscript; and

“The negative binding energies indicate that all the four proposed vdW heterostructures are structurally highly stable like the reported bilayer heterostructures [21,36]. The structure stability may imply the fabrication feasibility to some degree. As is known, BP/MoS2 [38], graphene/MoS2 [59], graphene/BP [39] bilayer heterostructures, and TMD/graphene/MoS2 [51] trilayer heterostructures have been successfully fabricated using physical and chemistry methods including mechanical stripping, liquid-phase stripping, hydrothermal and chemical vapor deposition. BlueP 2D monolayer has been well fabricated by molecular beam epitaxy [60], and the possibility to form its heterostructure with other 2D materials has been stated by a few studies [37,46]. Combining together the above fabrication abilities and considering the confirmed structure stability, the preparation of our proposed vdW heterostructures will be experimentally feasible. Noting the binding energy differences among these heterostructures, BlueP/BlueP/MoS2 and BlueP/graphene/MoS2 might be easier to be fabricated than BP/BP/MoS2 and BP/graphene/MoS2, although BlueP-related 2D materials seem currently farther away from practical manufacture than BP-related ones.”, to lines 185-199, pages 4 and 5 in the revised manuscript.

4) We are very sorry for our careless mistake, and it was changed from "Calculated TDOS and TDOS for ..." to "Calculated LDOS and TDOS for ..." in line 282 in the revised manuscript.

    We hope these revisions can address what the reviewer strongly concerns.

     Thank you very much for your concerning and review arranging.

Sincerely yours,

Hai-Zhi Song

Southwest Institute of Technical Physics

and

University of Electronic Science and Technology of China

Tel: +88-28-68180751

Email: hzsong1296@163.com,  hzsong@uestc.edu.cn

Round 2

Reviewer 1 Report

I appreciate the authors efforts to address my concerns. I recommend the manuscript for publication after further language revisions.

The manuscript reads well. It requires further very minor revisions at a few places. 

Reviewer 2 Report

The authors have properly addressed the reviewer's comments.